# Intensity of the Formation of Defects in Residential Buildings with Regards to Changes in Their Reliability

**Jarosław Konior**, **Marek Sawicki and Mariusz Szóstak** *

Department of Building Engineering, Faculty of Civil Engineering, Wroclaw University of Science and Technology, 50-370 Wrocław, Poland; jaroslaw.konior@pwr.edu.pl (J.K.); marek.sawicki@pwr.edu.pl (M.S.)

* Correspondence: mariusz.szostak@pwr.edu.pl; Tel.: +48-71-320-23-69

**Abstract:** Defining the basic determinants of the level of reliability with regards to the use of residential buildings and determining the function of the intensity of their characteristic defects are important issues concerning renovation strategy. The distribution of the exploitation time of residential buildings, the function of their reliability, and the distribution of the defect intensity of examined buildings are interdependent terms. Therefore, it can be assumed that the defect intensity of an object will be higher with an increase in its exploitation time. However, it is neither an increase reflecting the length of the building's service life nor the value directly proportional to its age. The article presents a model and method of testing the defects and reliability of a representative group of traditional downtown residential buildings, which were erected in Wroclaw, Poland at the turn of the 19th and 20th centuries. A basic conclusion was drawn regarding the mechanism of damage of residential buildings: for the period of using the facility, in which the time of correct operation until failure has an exponential distribution, the average remaining time of failure-free operation is unchanged at any time. It was confirmed that the tested residential buildings, after a certain period of failure-free operation, fulfil their functions, just like new buildings. The optimal moment of renovation occurs after the end of the second period of operation, before the period of rapid wear. The study of the course of the damage intensity function over time reflects the wear process of a residential building in a representative sample of downtown residential buildings that were erected using traditional methods. Defining the average duration of the correct failure-free operation of an object by the reliability function, which determines the probability with which the correct operation time of an object will be longer than its age, has a practical application in the exploitation of a residential building and its components.

**Keywords:** residential buildings; defects; intensity; reliability; technical wear

## 1. Introduction

### 1.1. Damage to Building Objects

Damage is an event that involves the loss of serviceability of an element or building [1–10]. It is related to them reaching their limit state. The exceeding of the limit state that is appropriate for the subsequent utility functions of individual elements of a residential building reduces their exploitation potential [11]. An element loses various utility functions when it reaches the serviceability limit state and enters the state that is defined in reliability theory as being defective (but fit for use). This state lasts until all of its functions exceed this limit state. The element then becomes unusable. The serviceability limit state is a contractual value, which depends on the adopted criteria.

The stimuli that cause the limit states to be exceeded by successive element functions may be sudden (random damage), gradual (aging damage), or have a nature of relaxation extortion (gradual aging of an element and its sudden transition to a state of being unfit for use occur together) [12]. Aging processes, which gradually occur, are usually caused by damage of a deterministic nature (predictable in a given time) [13]. Random damage occurs suddenly (breakdowns, catastrophes), or it is caused by accelerated wear (sporadic defects and technological defects) [14]. Defects (sporadic defects) are typical damage that result from the poor performance of executive works, the poor quality of the used construction materials, or they are caused by both of these causes simultaneously [15]. On the other hand, erroneous design assumptions and defective design and material solutions cause technological (chronic) defects are damage. Poor workmanship and the low-quality of built-in materials only exacerbate this problem [16].

If a technical element contains a sporadic or chronic defect, the limit state of its individual functions is reached faster. Subsequently, there is a clear reduction in the resistance of the material to external stimuli. Sudden damage causes unexpected changes in the essential physical parameters that determine the performance of the element's basic operational tasks. During a failure, the limit values of the safety functions of the element's structure are not exceeded, while, during a disaster, the parameters change beyond the values that are permissible by the requirements. Gradual damage is the result of aging activities. Aging processes are associated with irreversible structural changes in the materials that are used in building components. They result from physicochemical reactions that occur over time due to the operation of destructive stimuli on a macro and micro scale.

The rate of aging of materials depends on:

- the resistance of material to destructive stimuli; and,
- the intensity of the impact of destructive stimuli [17–21].

The accumulation of the effects of these interactions causes structural changes in material. The result of these external and internal destructive processes is the reduction of the material's resistance to damage that occurs during various periods of operation [22]. Consequently, there is a gradual increase in wear and a loss of functional properties of the element, which leads to its inoperability, and later to it being unfit for use. Exceeding the serviceability limit state by the individual operational functions of an element does not mean its full, physical destruction. Full physical destruction (and, at the same time, complete technical, social, and economic wear) occurs when the technical features of fundamental importance for the appropriate performance of the element's operational functions do not meet the parameters that guarantee safe operation [23]. The issue of safety is defined here by appropriate standards, technological guidelines, conditions of admission to use, approvals, and technical certifications [24]. Therefore, the criterion of safe operation is absolutely essential in the exploitation process, regardless of the nature of the causes of damage to a building's elements [25].

The so-called Lorenz curve illustrates the typical course of the wear process of building elements during their operation [26–28]. In this process, we can distinguish the following three basic intervals of a building's age t and the corresponding intervals of technical wear Zt—Figure 1:

- a warranty and post-warranty period of up to about 0.15 of a building's age t, in which the object "adjusts" and shows technical wear Zt at a level of 0.2,
- a period of normal exploitation of up to around 0.75 of a building's age t, in which the facility is properly maintained and shows technical wear Zt at a level of 0.5, and
- a period of planned exploitation of up to 1.0 of a building's age t, which is equal to its expected durability T, and in which the object should be renovated/modernized until it reaches a level of technical wear of 1.0.

The last period of planned exploitation and the period of unplanned use of residential buildings, the age of which exceeds their literature service life [29], is the subject of research and analysis regarding the intensity of damage and the change in the reliability state of buildings that qualified for the targeted research sample that is described below.

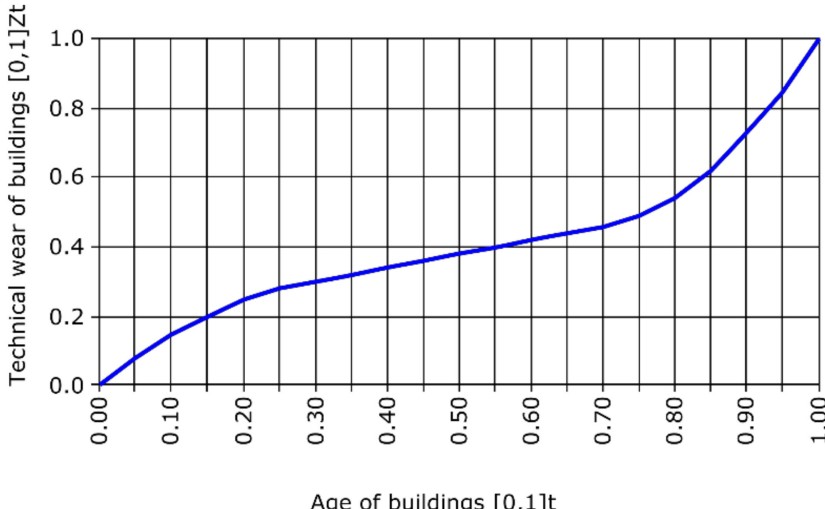

**Figure 1.** A typical course of the wear process of residential buildings during their exploitation.

*1.2. Literature Review*

During exploitation, construction objects are subjected to continuous destructive processes of various courses. With the passage of time, their functional properties decrease, and their partial restoration occurs as a result of repairs [30]. Therefore, during the use of buildings, it becomes necessary to carry out renovation works [31], which, according to the principle of sustainable development, should be included in the life cycle costs of construction objects [32–34].

Deciding which repair solutions to choose is a difficult and complex task. To this end, many models and methods have been developed in order to support policymakers and building administrators. They include the computer decision model for selecting repair options [35]; a simplified method of estimating technical degradation, which uses artificial neural networks [36]; and, the feasibility assessment of works using fuzzy stochastic networks [37].

In the proposed models and methods, an important element is the correct assessment of the size and intensity of defects to structural elements. A significant problem in the discussed issue is an increase of damage and partial defects [38], which is, the change in the building's reliability state during its operation. The analysis of the technical condition and intensity of damage requires appropriate modelling. The Rayleigh distribution [39,40] or Weilbull distribution [41,42] can be used for modeling this phenomenon.

The Weilbull exponential distribution is often used to evaluate the distribution of normal operation time [43], because it assumes that failures are only caused by external random events. However, in reality, there is no such exponential model of reliability distribution. Significant approximations, in which a negligible influence of wear processes is assumed, are made in the exponential distribution. A special example of the Weibull distribution is the Rayleigh distribution. This distribution, in turn, occurs when the wear of an element increases over time, i.e., it is the main cause of failure over time. The appropriate modelling of exploitation scenarios helps to select the optimal planning of renovation works for a building.

The aim of the research was to determine, while using the example of over 102 tested residential buildings, how the intensity of damage affects the reliability of construction objects.

## 2. Research Method

*2.1. Research Sample*

The subject of the research [44–48] involves tenement houses in a separate part of the downtown district in Wroclaw, Poland. The buildings are situated along downtown streets of secondary importance

in an urban layout that has remained unchanged for years. They are front buildings, and also outbuildings with a modest architectural design and economical functional standard. The facilities were built of brick in longitudinal, usually three-bay, structural systems.

102 tenement houses were mainly erected in the second half of the nineteenth century, until the outbreak of World War I. However, three of them are 170 years old. It is difficult to determine with certainty the type of building development due to the enormous scale of war damage that took place in this region in 1945; it can be assumed that, at the time of the examination, almost 2/3 of the buildings were built in compact developments, 1/5 in semi-compact developments, and 1/6 as free-standing buildings. The number of storeys varies from 2 to 5: 9% are two-story buildings, 10% are three-story buildings, 39% are four-story buildings, and 42% are five-story buildings. The vast majority of tenement houses (84%) have a basement under the entire building, 9% under a part of the building, and 7% have no basement at all. With the exception of three buildings, all of them have a usable attic. 83% of the attics are used as a drying room and 17% have been converted into apartments.

The apartments were designed without sanitary installations. Water intake points, as well as sinks and toilets (c.c.), were later installed on the staircase landings and even in the kitchens of the apartments. Furnaces heat most of the apartments, and only a few have central heating made by the residents themselves. Electrical installations, originally designed as surface-mounted, after the unprofessional modifications of tenants, are placed under the plaster. Gas installations were gradually introduced, with the development of the city network, to almost all apartments.

The term "tenement houses" defines the above-described downtown residential buildings with construction and material solutions that are typical for the turn of the 19th and 20th centuries, similar functions and standards, and a specific form of ownership (the so-called pre-war "tenement houses") in all parts of this article.

The research sample, covering 102 technically assessed residential buildings from Wroclaw's Srodmiescie district, was selected from a group of 160 examined buildings. The overriding criterion for the selection of the sample was the obtaining of a comparable group of objects. Mutual comparability of downtown tenement houses meant:

- age coherence, i.e., a similar period of erection, maintenance and exploitation with regards to historical and social aspects;
- compact development in the urban layout that has remained unchanged for years;
- similar location along downtown street routes with an urban, but not representative, character;
- construction and material homogeneity, especially regarding the load-bearing structure of buildings; and,
- identical functional solutions, which are understood as the standard of apartment amenities and furnishings in force at that time, and also a specific standard of living of residents.

A method of selecting the research sample at the level of greater detail was based on the mutual similarity of all technical solutions of downtown tenement houses.

The selected research sample, according to the criteria presented above, is representative with regards to one of the concepts (specific for the adopted purpose of the study) of representativeness [49,50]. It contains all the values of the variables, which could be recreated from previous research that had a different objective function than the one that was adopted in this study. However, these values were compiled and processed in such a way that it is possible to make conclusions about the cause–effect relationships between them in the general population. Thus, the typological representativeness of the sample into which the desired types of homogeneous variables are classified can be assumed. Because of the fact that the structure of the population and its properties were well recognized earlier, such a selection of the research sample can also be considered to be deliberate. It should be noted that the sample may not be representative in terms of the distributions of the examined variables, which may—for the adopted significance level—not correspond to the analogous distributions in the general population. It is also not known—at this stage of the research—whether the selected sample is

representative due to the correspondence between its variables and the identically defined variables in the entire set of downtown residential buildings.

### 2.2. Research Model

The general scheme of the cause-and-effect model—"defect–technical wear of building elements"—is the result of a synthesis of the results of visual studies of a selected sample of tenement houses in Wroclaw's Srodmiescie district. The scheme of the considered model at the level of greatest generalization is as follows:

$$[\text{CAUSES}] \rightarrow \left[\text{observed} \overset{\text{SYMPTOMS}}{\leftrightarrow} \text{measured}\right] \rightarrow [\text{EFFECTS}]$$

or in a more detailed elaboration—Figure 2:

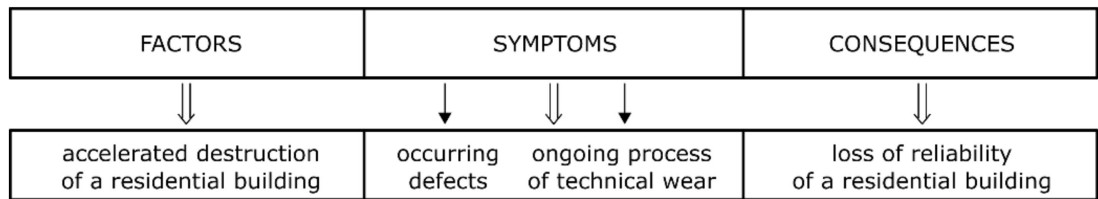

**Figure 2.** General diagram of the cause and effect model—"damage–technical wear of building elements".

The theoretical model of the technical wear of elements of residential buildings is a function of time t and their assumed durability T. The comparative analysis of the observed and theoretical wear shows that it is practically impossible to determine the exact form of the dependence between the size of wear of an element and its age. This difficulty results from the influence of many factors, which are individual for each residential building and can only be described by a complex mathematical model. In this situation, low complexity models should be selected and their compliance with empirical observations should be assumed as the selection criterion when the task of the researcher is to determine the trend of the phenomenon. Therefore, the research was limited to the search for trend functions from among linear, power (multiplicative), as well as exponential and hyperbolic relationships [51–57].

At any time during the assessment of the technical condition of any selected group of residential buildings and their elements, the group of experts acts at the intermediate stage of the proposed model—the analysis of the symptoms of the observed states. It cannot measure the causes (factors). However, it may take their impact into account when carrying out the assessment. In the case of effects (consequences), short-term effects (e.g., loss of utility values) and intentional effects (concerning decisions about the future of a residential building) can be distinguished. Further steps depend on adopting one of the multi-criteria decision making methods, e.g., according to [58]. The more reliable and meaningful the research on the symptoms of damage to a residential building's elements in the observed states, the more reliable the reasons regarding further decision-making analysis.

The key element of the technical examination of residential buildings should be the assessment of the size and intensity of the damage (symptoms) of their structural elements, which is carefully prepared in advance in terms of methodology. This assessment, which is supported by the theoretical recognition of the failure mechanism with regards to the reliability of technical facilities, leads to the determination of the causes of damage, and it enables a decision that is based on numerical evidence concerning the future of residential buildings to be made.

### 2.3. Research Method

The concept of the reliability of a residential building is always associated with the performance of exploitation tasks [7,59–63]. The performance of a task by a residential building involves its correct fulfilment of certain functions under certain operating conditions and within a specified time. If this

function is denoted by φ, the building's working conditions by χ, and the building's operation time by t, then the task to be performed by the facility can be written as an ordered triple [φ,χ,t]. By knowing the function that the building needs to perform, it t is possible to establish such a set of requirements (ωφ) for the features of a residential building (characterized by a number of essential and auxiliary technical-operational, economic, and other parameters that are important in the process of exploitation and maintenance) that their fulfilment is a necessary and sufficient condition for the correct implementation of the assigned functions (φ) by the building. It has been assumed, with some simplifications, that the assessed residential building is, from the point of view of exploitation theory, a two-state object. This means that it may be fit to perform its function (assuming the actual state that is characterized by meeting the requirements (ωφ)), or unfit to perform its function (assuming the physical state that is characterized by a failure to meet the requirements (ωφ)). The task of the facility, which is then understood as an event (Z) (e.g., with regards to the provision of housing services), is written as the following ordered triple: [ωφ,χ,t]. It was further assumed that the requirements regarding a residential building and its maintenance conditions are known, i.e., the pair [ωφ,χ] is fixed and, consequently, it was assumed that the reliability of residential buildings could be assessed as a function of time (t).

Thus, the concept of the reliability of a residential building is defined, as follows: the reliability of a residential building is its property, which is seen as its ability to meet the requirements (ωφ) within the designated limits of being fit and unfit under certain given maintenance conditions (χ) and exploitation time (t).

The above considerations allowed for the reliability measure to be defined according to general formula [30,61]:

$$R(t) = P\{\tau > t\} \tag{1}$$

where:

$\tau$—time of the failure-free operation of an element; and,

*R(t)*—the function of reliability that describes the probability of the failure-free operation of an element during the time period *t* (it assumes values from the interval [0, 1], where *R(0) = 1*, and *R(∞) = 0*), which can be also expressed as *lim{t -> ∞} R(t) = 0*.

More precisely, the reliability function R(t) denotes the probability of the correct operation of an object in the interval [0, *t*]. For the considered residential buildings with repairable elements, the index (1) characterizes their reliability until the first defect. The course of the reliability curve coincides with the change in the serviceability value due to the fact that they are also complex objects.

The function *R(t)* is also a transformed distribution function, the form of which is as follows:

$$F(t) = P\{\tau > t\} = 1 - R(t) \tag{2}$$

*F(t)* determines the probability with which the time of the correct operation of a residential building will be shorter than its expected service life (*t*). If it is assumed that such an event occurs at time *t = 0*, i.e., an object is fit for use at the moment of putting it into operation, then it can be assumed that it will also perform this task at any time $t_i$ (from interval *0<$t_i$ ≤ t*, where *i = 1,2, . . .* ). The general reliability of a residential building is then equal to the product of reliability:

$$R = R(t)R(0) = R(\omega\varphi, \chi, t)\Delta P\{Z(\omega\varphi, \chi, 0)\} \tag{3}$$

The symbol *R(0)* denotes the so-called initial reliability of an element, i.e., the probability that it will be fit at the moment of starting the task *(t = 0)*. Therefore, the definition of the concept of reliability was used and, as a result, the following was obtained:

$$R(0) = P\{Z(\omega\varphi, \chi, t)\} \tag{4}$$

and then:

$$R(t) = P\{Z(\omega\varphi, \chi, t_i)\},\ 0 < t_i < t/Z(\omega\varphi, \chi, 0) \tag{5}$$

It was further assumed that event $Z(\omega\phi, \chi, 0)$ is certain, i.e., it is certain that, if a residential building adopted from the investment process to the exploitation process meets all the specified functions, then it will fulfil all the operational tasks assigned to it and it can also be inhabited by residents. In this case, $R(0) = 1$, and expression (5) takes the following form:

$$R(t) = P\{Z(\omega\varphi, \chi, t_i)\},\ \ 0 < t_i \le t \tag{6}$$

Ultimately, the overall reliability of a residential building can be expressed by the $R(t)$ function, which may have different distributions in different periods of the building's operation. Most often, variable ($t$) can be treated as a random variable of the continuous type and then the density function is a derivative of the distribution function $F(t)$:

$$f(t) = F'(t) = -R'(t) \tag{7}$$

and then:

$$F(t) = \int_0^\infty f(t)dt = \int_0^\infty \left(-R'^{(t)}dt\right) \tag{8}$$

These are important relationships in the process of analysing building structures.

Note that the reliability function is a decreasing function. This means that e.g., $R(t_i) < R(t_{i-1})$, and, in extreme cases for $t_i = 0$ and $t_i = \infty$, the reliability function takes the values $R(0) = 1$ and $R(\infty) = 0$. This is contrary to the distribution function F(t), also called the unreliability function, for which $t_i$: $F(0) = 0$ and $F(\infty) = 1$ for the same values.

Further considerations were based on the definition of durability ($T$), which can be formalized as a function of the following quantities [61,64]: the reliability of a residential building $R(t)$, which is considered to be an event of randomly reaching the limit state by its element; the flux of physical aging extortions of a building and its elements $W(t)$ (sometimes in the form of step stimuli); and, the level of resistance of elements to the effects of extortions $D(t)$. When presenting the problem with regards to reliability, it can be defined as the functional:

$$R(t) = \Psi\{T(t), W(t), D(t)\} \tag{9}$$

in which reliability is expressed by the durability and flux of changes in the level of aging and, thus, the processes of physical wear (if the problem is simplified by not taking the processes of social wear into account). The wear processes are manifested by damage to a residential building and its elements, as a result of which the building is unfit for use (loses its serviceability value) and requires renovation (if it is technically possible and economically justified). With exploitation time, a residential building becomes unfit for use more often, as it is subjected to increasingly frequent damage (aging processes). For the sake of simplifying the considerations, distinguishing between the state of operability and inoperability was omitted. However, the subsequent state of being fit for use, after renovation, already represents a level of serviceability that is lower than the previous one. Therefore, it is possible to talk about a greater intensity of damage to the object with an increase in its operation time, although this is neither an increase reflecting the length of the building's service life nor an increase directly proportional to its age.

An important issue in renovation strategy is to define the basic determinants of the level of reliability concerning the use of residential buildings. The basic element of these studies is the determination of the defect intensity function, which is itself an important issue with regards to the assessment of renovation decisions regarding residential buildings.

When considering any two renovation intervals, it can be assumed that the random variable $\tau$, which defines the time of failure-free operation of a building, takes values from the following interval [61]:

$$[\tau_{Ri} < \tau \leq \tau_{Ri+1}] \tag{10}$$

or, in general:

$$[t_i < \tau \leq t_i + \Delta t_i] \tag{11}$$

If it is assumed that in the interval $[0,t_i]$ no damage was found and, in the interval $[t_i,t_i+\Delta t_i]$, damage could occur, the expression $R(t_i+\Delta t_i)/R(t_i)$ is called the conditional probability of such an event, which assumes that there will be no damage in the interval $[t_i+\Delta t_i]$ if there is no damage in the interval $[0,t_i]$. It is a relationship with a domain defined on the interval $[0,1]$, also known as the Bayesian formula concerning conditional probability [37,48]:

$$P\{t_i; t_i + \Delta t_i\} = \frac{R(t_i + \Delta t_i)}{R(t_i)} \tag{12}$$

In extreme cases, damage may occur at the moment of $t_i+\Delta t_i$, and then the probability that is determined by equation (12) will assume value 1 (because $R(t_i+\Delta t_i) = R(t_i)$). If the damage occurs at the moment of $t_i$, then $P\{t_i;t_i+\Delta t_i\} = 0$ (and then $R(t_i+\Delta t_i) = 0$). Formula (12) indicates the conditional probability of the duration of the correct and fault-free operation of the object.

In subsequent steps, the following transformations were made, which allowed the concepts derived from expression (12) to be defined:

- relationship (12) was subtracted from unity:

$$1 - P\{t_i; t_i + \Delta t_i\} = U\{t_i; t_i + \Delta t_i\} \tag{13}$$

and the resulting complement $U\{t_i;t_i+\Delta t_i\}$ was interpreted as the probability of a faulty and incorrect operation of the object;

- the obtained expression (13) was divided by $\Delta t_i$, and the average value of the probability of damage in the object's operating time interval with the length $\Delta t_i$ was obtained:

$$\frac{U\{t_i; t_i + \Delta t_i\}}{\Delta t_i} \tag{14}$$

- the limit (probability) of this transformed expression for $\Delta t_i \to 0$ was then calculated and the sought function of damage intensity $\lambda(t_i) = \lambda(t)$ was obtained:

$$\lim_{\Delta t_i \to 0} \frac{U\{t_i; t_i + \Delta t_i\}}{\Delta t_i} = \lambda(t) \tag{15}$$

- the obtained relationship (15) was further transformed through appropriate integration and differentiation operations until a more convenient mathematical form was obtained. It expressed the relationship between the function of damage intensity and reliability function:

$$R(t) = e^{-\int_0^t \lambda(t)dt} \tag{16}$$

Relationship (16) is the basic formula in the theory of exploitation, which is used under the name of the Wiener formula.

The distribution of the service life (exploitation) of a residential object $f(t)$, the reliability function $R(t)$, and the distribution of damage intensity $\lambda(t)$ are interdependent terms. The function of damage

intensity itself depends—just like the reliability function (because it is a set of its arguments)—on many factors. These include the physical and chemical properties of the elements of residential buildings, the aging and wear processes $W(t)$, the requirements for these objects ($\omega\phi$), and the maintenance conditions of these objects ($\chi$). The damage intensity function $\lambda(t)$ can take various forms. It may be monotonically increasing or decreasing (possibly with a few extreme points) or it may be constant over time. The time course of the damage intensity function $\lambda(t)$ reflects the course of the wear processes of a residential building throughout its service life. Table 1 shows the average values of the damage intensity function $\lambda(t)$ for the ten most important elements of the examined downtown tenement houses, as well as the indication of the lack of damage intensity $\lambda(t) < 0.12$, the damage tendency $0.12 < \lambda(t) < 0.20$, and a strong damage intensity $\lambda(t) > 0.20$.

The statistical form of this reliability measure is also used apart from the probabilistic (based on the probability calculus) approach for determining the failure intensity function in the exploitation theory [61]:

$$\overline{\lambda(t)} = \frac{n(t + \Delta t) - n(t)}{N(t)\Delta t} \tag{17}$$

where:

$N(t)$—the number of objects fit for use until time $t$;

$N(t + \Delta t)$—the number of damaged objects until time $t + \Delta t$; and,

$\overline{\lambda(t)}$—a statistical measure of damage intensity.

$\overline{\lambda(t)}$ is therefore the share of the number of defects in the analysed time unit $(\Delta t)$ in interval $[t, t+\Delta t]$ and in the number of objects fit for use at the beginning of this interval, i.e., at time $t$.

For the renovation strategy of a residential building, an important issue is to find—with the assumed reliability level—a rational moment of renovation $\tau R$, i.e., to determine the most technically and economically advantageous average time of exploitation of the object $\tau 0$ from one renovation to the next one. The measure of the average inter-repair time (or the value of the average random variable $\tau$, i.e., the variable that determines the time of the correct operation of the facility) is:

$$\tau_0 = E(\tau) = \int_0^{\tau R} R(t)dt \tag{18}$$

It should be assumed that, in the period of building adaptation, i.e., for objects damaged during this period, the distribution of the damage intensity function is usually explained by various functions that have a monotonic and decreasing character of their course (more or less, depending on the nature of the damage intensity). The processes of the adaptation period are associated with the loss of serviceability, which is caused by the exceeding of the limit states. After a fairly long-term effect of loads, when they are continuously distributed and cyclically repeated in the process of exploitation, the Weibull function can be used, the density of which has the form of:

$$f(t) = \frac{\beta}{\alpha}t^{\beta-1}e^{\frac{t\beta}{\alpha}} \tag{19}$$

where: $\beta$—shape parameter, $\alpha$—scale parameter.

Because the damage intensity function can be written using the parameters $\alpha$ and $\beta$ in the form:

$$\lambda(t) = \frac{\beta}{\alpha}t^{\beta-1} \tag{20}$$

then:

$$R(t) = e^{-\int_0^{\infty} \frac{\beta}{\alpha}t^{\beta-1}dt} \tag{21}$$

**Table 1.** Average values of the damage intensity function λ(t) for the ten most important elements of residential buildings.

| defect No | type of defect | Foundations λ(t)2 | Walls of Basement λ(t)3 | Solid Floor over Basement λ(t)4 | Main Walls λ(t)5 | Inter - Storey Wooden Floors λ(t)6 | Stairs λ(t)7 | Roof (Rafter Framing) λ(t)8 | Window Joinery λ(t)9 | Inner Plasters λ(t)10 | Facades λ(t)11 |
|---|---|---|---|---|---|---|---|---|---|---|---|
| d1 | Mechanical defects | | | | | | 0.05 | | **0.29** | 0.09 | **0.28** |
| d2 | Leaks | | | | | | | | **0.26** | | |
| d3 | Mechanical decrements of bricks | 0.13 | **0.23** | 0.08 | **0.19** | | 0.03 | | | | |
| d4 | Mechanical decrements of mortar | | **0.28** | | **0.30** | | | | | | |
| d5 | Decrements caused by rotten bricks | 0.14 | 0.07 | 0.00 | 0.17 | | | | | | |
| d6 | Decrements caused by rotten mortar | | 0.05 | | 0.09 | | | | | **0.47** | **0.48** |
| d7 | Paint coating's peeling off | | | | | | | | | 0.15 | **0.55** |
| d8 | Paint coating's falling off | | | | | | | | | **0.25** | **0.57** |
| d9 | Craks of bricks | 0.05 | 0.01 | 0.05 | 0.11 | | | | | | |
| d10 | Craks of plaster | | 0.03 | | 0.03 | | | | | **0.30** | **0.63** |
| d11 | Scratching of walls | | | | **0.21** | | | | | | |
| d12 | Scratching of plaster | | | | 0.12 | 0.05 | | | | **0.18** | **0.63** |
| d13 | Loosening of plaster | | | | | 0.09 | | | | **0.67** | **0.81** |
| d14 | Plaster's falling off | | | | | | | | | **0.57** | **0.50** |
| d15 | Signs of permanent damp | **0.70** | **0.74** | **0.58** | **0.56** | 0.07 | | **0.43** | **0.83** | **0.70** | **0.84** |
| d16 | Weeping | **0.64** | **0.52** | **0.67** | **0.46** | **0.27** | **0.59** | **0.50** | **0.74** | **0.61** | **0.79** |
| d17 | Biological corrosion of bricks | **0.36** | **0.31** | | **0.67** | | | | | | |
| d18 | House fungus | | | | | **0.45** | | | | **0.38** | **0.60** |
| d19 | Mould & decay | **0.49** | **0.43** | | **0.34** | | | | **0.49** | **0.41** | **0.56** |
| d20 | Localized corrosion of steel beams | | | **0.42** | | | **0.54** | | | | |
| d21 | Surface corrosion of steel beams | | | **0.29** | | | **0.61** | | | | |

**Table 1.** *Cont.*

| | | | | | | | | | | | |
|---|---|---|---|---|---|---|---|---|---|---|---|
| d22 | Pitting corrosion of steel beams | | | **0.55** | | | | 0.53 | | | | |
| d23 | Flooding of foundation | | | **0.45** | | | | | | | | |
| d24 | Wooden beams of floor sensitiveness to dynamic activity of human's weight | | | | | 0.00 | | | | | | |
| d25 | Deformation of wooden beams | | | | | 0.12 | | | | | | |
| d26 | Skewing of joinery | | | | | | | | | **0.42** | | |
| d27 | Warp of joinery | | | | | | | | | 0.04 | | |
| d28 | Stratification of wooden elements | | | | | | | 0.07 | | | | |
| d29 | Partial deterioration of wooden elements pest attacked | | | | | | 0.38 | 0.28 | 0.45 | | | |
| d30 | Total deterioration of wooden elements pest attacked | | | | | 0,43 | | 0.57 | 0.42 | | | |
| | number of cases: | 100 | 93 | 93 | 100 | 100 | 100 | 100 | 100 | 97 | 100 | |

Damage intensity function λ(t) appears in three following ranges: λ(t)—lack of intensity defects formation; **λ(t)**—tendency to intensity defects formation; **λ(t)**—strong intensity defects formation.

A typical distribution for objects subjected to damage in the first period of exploitation may be the gamma distribution in the form:

$$f(t) = \frac{\lambda(\lambda(t))^{\beta-1}e^{-\lambda(t)}}{\int_0^\infty e^{-t}t^{\beta-1}dt} \tag{22}$$

in which the denominator is the so-called Euler's integral. This distribution represents the processes of loss of serviceability well, which are based on the cumulative effects of external factors. Ultimately, as a result of the elimination of all the defects and faults covered by the warranty, it is assumed that gradual and sudden defects (the latter ones that result from the step operation of stimuli in the conditions of accumulation of wear) are completely eliminated. This characterizes the end of the adaptation period of a residential building and the beginning of its normal operation period. The shape of the intensity function becomes "smooth". Hence, value 1 for the shape parameter ($\beta$) can be assumed for both the Weibull distribution and gamma distribution. It is a very characteristic period of a residential building's service life, in which the risk function takes a constant value ($\lambda(t) = \lambda$). This results from the following transformations of the mentioned functions:

- for the Weibull function:

$$\lambda(t) = \frac{\beta}{\alpha}t^{\beta-1} = \frac{1}{\alpha}t^{1-1} = \frac{1}{\alpha} = \lambda \tag{23}$$

- for the gamma function (using a shortcut for long calculations):

$$\lambda(t) = \frac{f(t)}{R(t)} \tag{24}$$

$$f(t) = \lambda e^{-\lambda t} \tag{25}$$

$$F(t) = 1 - e^{-\lambda t} \tag{26}$$

and using dependence (2):

$$R(t) = e^{-\lambda t} \tag{27}$$

ultimately:

$$\lambda(t) = \frac{\lambda e^{-\lambda t}}{e^{-\lambda t}} = \lambda (= const) \tag{28}$$

If the risk function has a domain defined by segment ti of the exploitation time ($0 < t_i \leq t$), then $\lambda(t)$ = const (for $\lambda > 0$ and $t > 0$) and the time of correct operation of a residential building has an exponential distribution. Finally, after several mathematical transformations of Equation (18), it is possible to determine the most technically and economically advantageous average service life of the object $\tau0$ from the end of the warranty period:

$$\tau_0 = \frac{1}{\lambda} \tag{29}$$

The last dependence leads to an extremely important conclusion for the mechanism of the occurrence of defects in residential buildings: for the period of using a facility, in which the time of correct operation to damage has an exponential distribution, the average remaining time of failure-free operation is unchanged at any time. Therefore, after a certain time of failure-free operation, residential buildings fulfil their functions, just like new ones, and after exceeding the planned exploitation time beyond the assumed service life ($t > T$), they show "over-durability"—Figure 3:

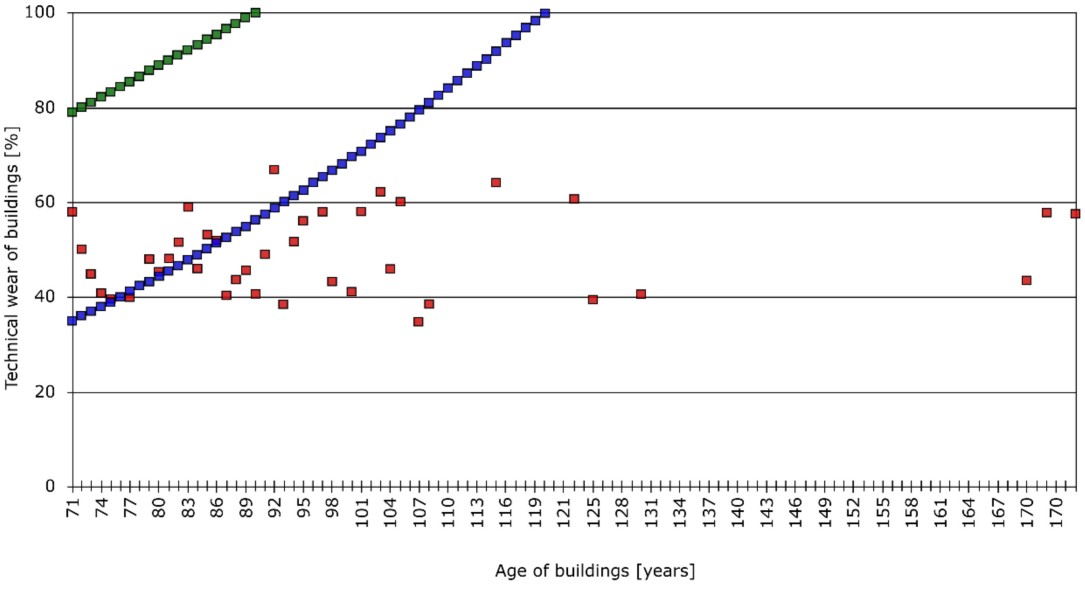

**Figure 3.** A diagram of the identified "over-durability" of the examined downtown tenement houses with regards to the charts of theoretical technical wear during very good and bad maintenance conditions.

It is clearly seen in Figure 3 that the observed technical wear of examined buildings is within the area that is determined by theoretical wear at poor and good maintenance by the age of 85 years. Within the range of $85 < t < 120$, the theoretical curves do not match the observed ones (the older, the worse). Over the age of 120 years, the inspected buildings should head for "technical death" ($t = T$), whereas a trend of "over-durability" ($t > T$) is noted on regular basis.

## 3. Conclusions

The adopted model and method of testing a representative group of downtown residential buildings with a traditional construction, which were erected at the turn of the 19th and 20th centuries, indicate that the age of the elements of old residential buildings is of secondary importance in the process of the intensity of loss of their serviceability value. No more than 30% of the element's damage can be explained by the passage of time if we assume that the coefficient of determination is the measure of the adjustment of the mathematical models (as a function of the technical wear of building elements over time), which are tested in the nonlinear regression method. Therefore, it is not age that determines the course of the technical wear of the analysed building components.

The analysis of the exploitation processes of residential buildings and the transformations of the basic dependencies of the reliability theory indicate that, for the service life of an object, in which the time of correct operation to failure has an exponential distribution (it is basically the service life corresponding to the length of operation of the considered residential buildings), the average remaining time of failure-free operation is unchanged at any time. Theoretically, residential buildings fulfil their functions, just like new ones, after a certain period of failure-free operation. The optimal moment of renovation occurs after the end of the second period of operation, before the period of rapid wear. Expressing the average duration of the correct failure-free operation of an object (τ0) by the reliability function *R(t)*, which determines the probability with which the correct operation time of an object will be longer than $t_i$, has a practical application in the exploitation of a residential building and its components.

The study of the course of the damage intensity function $\lambda(t)$ over time reflects the wear process of a residential building in a representative sample of downtown residential buildings erected while using traditional methods. It authorizes the formulation of the following conclusions—Table 1:

- there is measurable the damage intensity function in interval [0, *t*] for all 10 tested building elements, but the damage intensity force shows a significant span (from 0.00 to 0.84);
- as a rule, damage that is caused by water penetration and moisture penetration is of the highest intensity -0.54 on average;
- the technical condition of each of the tested elements also shows the intensity of defects that are characteristic for their design and material solutions, e.g.,:

  ○ damage to wooden parts of elements (ceiling beams, stair treads, roof trusses, window joinery), which are attacked by biological pests;
  ○ mechanical damage to the structure and texture, the intensity of which applies only to those elements in which the damage may cause the intensification of the impact of subsequent (cumulative) defects, e.g., construction walls underground and aboveground, as well as internal and external plasters (but not foundations or massive cellar ceilings); and,

- damage that is manifested by the loss of the original shape of wooden elements can be considered as not very intense; an exception is the torsion of window joinery (with an intensity of 0.42), for which this damage determines a significant decrease in its serviceability value.

## 4. Summary and Discussion

The practical approach to the problem of the intensity of the formation of defects in the tested residential buildings with regards to the change in their reliability condition enables summing up the following findings:

- generally used normative definitions of the reliability of buildings facilitate the study and interpretation of the course of exploitation processes of residential buildings;
- for the purpose of a comprehensive assessment of changes in the reliability level of residential buildings, various reliability characteristics should be used, in which the damage intensity function is of key importance, as it enables the construction of other reliability indicators; and,
- using the characteristics of the reliability of a residential building in renovation decisions allows for a rational renovation strategy to be determined by e.g., the determination of maintenance intervals on the basis of established damage intensity distributions.

The methodological approach to the technical assessment of buildings, and their durability and reliability, has been known and presented in the literature for many years, especially in the papers of Arendarski [5], Zaleski [8,9], Thierry [21], and Tymiński [61]. Their works are used as manuals for managers and administrators regarding the use of buildings. Nowogońska [11,18,28,30,31,39,40] deals with the study of the impact of the maintenance of residential buildings on both the degree of damage and the reliability function. The reliable results of these studies are presented with a division into building elements with the greatest share and significance for the proper functioning of the examined buildings. Such a division is particularly important in the last period of the building's "service life", when its operation time is approaching its expected durability.

A similar approach, as presented in previous publications, was adopted by the authors of this article, who conducted the technical assessment of tenement houses under the supervision of Marcinkowska and Czapliński [44–48,58]. This methodology of diagnosing residential buildings has been presented for many years by the researchers of the so-called "German school"—Deutschmann [6] and Zimmermann [10]. They indicated the methods of measuring technical wear, damage size and the aging process of the load-bearing structure of engineering objects. Plebankiewicz, Zima, and Wieczorek [32,33] deal with the life cycle of a building object with regards to the risk and cost

of its restoration as a result of renovation activities, which—in the opinion of the authors of this publication—should be a secondary feature: cost versus technical. After all, increasing cultural, historical, and humanistic aspects prevail over material ones when making decisions about the so-called "technical death" of facilities that are located in the centres and suburbs of many cities. Therefore, the priority seems to be to study the damage intensity of residential buildings in the context of changes in their reliability state.

It is worth noting that the discussed quantitative data can provide the basis for programming the size and structure of specialized construction companies that are involved in the maintenance and renovation of residential buildings. These data are included in the technical information that is necessary for managing buildings and designing the organization of these maintenance activities for residential buildings, which, in turn, determines the quality of broadly understood housing maintenance conditions.

Finally, attention should be paid to the individual nature of the results of the study, which was based on research on a homogeneous coherent group of downtown tenement houses. The transfer of the results of the technical assessment to a different population of residential buildings with a traditional construction should be conducted with great caution and with the necessity to perform surveys. Undoubtedly, such studies should be preceded by the careful, purposeful selection of a typological sample that is representative for the general population. Such a sample may contain a much smaller number of objects, but it is extremely important that the decisive selection criterion for the technical assessment involves the elements (or only parts of them) that are essential for the structure (load-bearing structure) of the building. This division is especially important when examining composite and complex elements. It can then be assumed that, from the point of view of the ultimate limit states of elements, the degree of their technical wear (while maintaining the safe and reliable operating conditions of the object) is equal to 75%.

The methodological aspects of the reliability of the quantitative results of the technical assessment should also aim to minimize the subjectivity of expert judgment in the process of technical examinations of residential buildings by specifying the type of the predicted random impacts, and by determining the variability of at least some of them. It should also be remembered that the issue of technical tests of buildings (especially residential buildings) needs to be updated with full recognition of the forms of their immaterial wear—social and economic. It is a sign of recent times that it is the psychological aspects of the perception of the process of decline in the serviceability value of flats by their users, being supported by the analysis of the profitability of replacing entire buildings, which play a fundamental role when making decisions regarding the future of downtown housing developments.

Therefore, the results of the research should be treated as an exploratory study, the main aim of which was to model the solution of cause–effect relationships. These dependencies indicate the type and size of damage that shows the impact of the maintenance conditions of downtown residential buildings on the technical wear of their components. Like any exploratory solution, it should be treated as a multi-criteria recognition of the mechanism of the occurrence and effects of phenomena that are encountered by an adjudicator at each stage of the technical assessment of an engineering object. However, this assessment, in its nature, includes an unmeasurable (partly subjective) aspect. The construction of a new model of the technical inspection of residential buildings, which is based on the assumptions and conclusions resulting from the study, will allow the burden of the results of the technical assessment to be considered as more quantitative than qualitative. The intention of the authors is that further work related to the broadly understood diagnosis of technical objects should go in this direction.

**Author Contributions:** Conceptualization, J.K., M.S. (Marek Sawicki) and M.S. (Mariusz Szóstak); methodology, J.K.; software, J.K. and M.S. (Mariusz Szóstak); validation, J.K., M.S. (Marek Sawicki) and M.S. (Mariusz Szóstak); formal analysis, J.K., M.S. (Marek Sawicki) and M.S. (Mariusz Szóstak); investigation, J.K. and M.S. (Marek Sawicki); resources, J.K. and M.S. (Marek Sawicki); writing—original draft preparation, J.K.; writing—review and editing, J.K. and M.S. (Mariusz Szóstak); supervision, M.S. (Marek Sawicki). All authors have read and agreed to the published version of the manuscript.

**Funding:** This research received no external funding.

**Conflicts of Interest:** The authors declare no conflict of interest.

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
