# Peer review of "Intensity of the Formation of Defects in Residential Buildings with Regards to Changes in Their Reliability"

_applsci, doi:10.3390/app10196651_

Round 1
Reviewer 1 Report
Detail Comments on the article “Intensity of the Formation of Defects in Residential Buildings with Regards to Changes in Their Reliability” Applied Sciences
General Overview:
This paper investigates the basic determinants of the level of reliability with regards to the use of residential buildings, and to determine the function of the intensity of their characteristic defects.
However, the novelty of the work is marginal, hence serious improvement is needed.
Also, the following issues are needed to be considered before it can be considered for publishing.
- Abstract needs to be precise and clearly explained i.e. findings.
- Authors need provide a clear view of Table 1. The table is not legible and better arrangement is necessary.
- Figure 3 needs more explanation with references e.g. Eq. (…).
- Also the results presented in Figure 3 doesn’t agree with the typical pattern shown in Figure1.
- Conclusion is also not convincing for the readers; it is too long and not clearly justified.
Author Response
Dear Mr. Kongpat Jirapisaisuk, Assistant Editor – MDPI,
Dear Editorial Board of Applied Sciences,
Thank you for the review of our paper entitled “Intensity of the Formation of Defects in Residential Buildings with Regards to Changes in Their Reliability” to be published in the Journal of “Applied Sciences”.
First of all, we appreciate your thoughtful and accurate comments presented by the reviewers. We have carefully considered all remarks and have now completed the revisions incorporating their suggestions in the revised manuscript (copy attached).
We hope that we have taken all your critical and constructive comments into account in the revised paper. We also hope that the current version meets your expectations.
Kind regards,
Jarosław Konior, Marek Sawicki and Mariusz Szóstak
Department of Building Engineering, Faculty of Civil Engineering, Wroclaw University of Science and Technology, 50-370 Wrocław, Poland
Here are answers to reviewer’s comments – one by one:
Detail Comments on the article “Intensity of the Formation of Defects in Residential Buildings with Regards to Changes in Their Reliability” Applied Sciences
General Overview:
This paper investigates the basic determinants of the level of reliability with regards to the use of residential buildings, and to determine the function of the intensity of their characteristic defects.
However, the novelty of the work is marginal, hence serious improvement is needed.
Also, the following issues are needed to be considered before it can be considered for publishing.
Comment 1. Abstract needs to be precise and clearly explained i.e. findings.
Answer 1. The abstract has been supplemented by accurate findings linked with drawn conclusions in terms of the course of the damage intensity function λ(t) over time and the reliability function R(t), which determines the probability with which the correct operation time of an object will be longer than its age.
Comment 2. Authors need provide a clear view of Table 1. The table is not legible and better arrangement is necessary
Answer 2. Table 1 has been improved. We hope that in the current version is readable.
Comment 3. Figure 3 needs more explanation with references e.g. Eq. (…).
Answer 3. Explenation of figure 3 and its meaning inserted below: It is clearly seen in the figure 3 that the observed technical wear of examined buildings is within the area determined by theoretical wear at poor and good maintenance by the age of 85 years. Within the range of 85 < t < 120 the theoretical curves do not match the observed ones (the older, the worse). Over the age of 120 years the inspected buildings should head for “technical death” (t = T) whereas a trend of “over-durability” (t > T) is noted on regular basis.
Comment 4. Also the results presented in Figure 3 doesn’t agree with the typical pattern shown in Figure1.
Answer 4. This is the point! See the answer above. Please note that results in the figure 3 are presented from the of 71 years where theoretical formulas do not work and do not represent the state of reality. Both damage intensity function λ(t) over time and the reliability function R(t) with which the correct operation time of an object is longer than its age explain the reasons.
Comment 5. Conclusion is also not convincing for the readers; it is too long and not clearly justified.
Answer 5. The conclusions have been simplified and linked directly to the research method described in item 2.3 with numerical results presented in the table 1. Soft / general conclusions have been reformulated and moved to summary in item 4.

Reviewer 2 Report
The authors performed an exploratory study on the formation of defects by modelling the reliability function of buildings. In particular, the proposing model is tested against a representative group of residential buildings with regards to technical wear. The study used a probabilistic approach, based on the Bayesian conditional variable framework, to model the distribution of damage intensity function. Such a function considered multiple factors such as the building's fulfilment functions and working conditions, as the technical wear of a building is naturally affected by multiple criteria. In addition, a statistical form of the reliability measure is also presented. The study has drawn the conclusion of age not being the main factor that determines the course of the technical wear. Overall, the result of the paper presented a methodology that was able to model the cause-effect relationships of damage to residential buildings.
The problem that this paper address is grounded and the proposed method is logically sound. Moreover, the proposed method addresses the issue with a probabilistic approach in modelling the failure intensity function. The study of the damage intensity function allows one to assess the wear process based on samples obtained from residential buildings.
The following are some comments on the formulation of the proposing method:
- In line 237, the reliability function R(t) is said to "assumes values ... where R(0) = 1, and R(\infty) = 0". The statement would be more appropriate to be written as the limit of: lim_{t->\infty} R(t) = 0.
- It is stated in line 237 that R(0) = 1. However, the subsequent lines in 246-248 then begins to generalise and define that R(0) is the initial reliability where eq3 stated R(0) = P{Z(\omega\varphi,X,t)}. It is not until line 254 that an additional assumption is stated of saying the event Z(\omega\varphi,X,t) is assumed to be certain and hence R(0) = 1. Perhaps the ordering of formulation should be rearranged.
- In line 240 (and 410), the phrase "right-hand time interval" implies there is an intrinsic direction of positive and negative numbers, which is not true—referring the interval as the positive direction would be sufficient.
- Eq 2 seems to be incorrect. Based on eq1 and the text descrption, perheps it was supposed to be: F(t) = P{ \tau <= t } = 1 - R(t)?
- In some instances ti (without subscript) is used (e.g. line 251) while in some other instance, t_i (as a subscript) is used (e.g. eq5, line265)
- R(t) takes time as the input. But in eq 4, the reliability function R(.) is the first instance of R where it takes the tuple of (\omega\varphi,X,t). Was it meant to be P{Z(\omega\varphi,X,t)}?
- Eq8 has an open parenthesis but with no closing parenthesis
- Given the definition of inequality of t_i in line 251 and in eq6, i.e. 0 < t_i <= t, the statement in line 265 is invalid "in extreme cases for t_i=0 and t_i=\infty". t_i does not takes the value of 0.
- In line 296, the inverval does not seems correct. Was it meant to be "....in the interval [t_i, t_i + \delta t_i]....."?
- In addition to the last comment, does it matter to the analysis if the interval is an open or closed interval (since the two intervals coincide at t_i)?
- Eq12. It is confusing and unclear how the authors derive eq 12 when R is being reformulated under the Bayesian conditional probability. Firstly, with conditional probability, P(A|B) is normally used to denote the probability of A given B. Moreover, the classical Bayes' theorem on conditional probability is in the form of P(A|B) = P(B|A) P(A) / P(B). It is not immediate to the reviewer how eq12 is derived under the Bayes' theorem.
- Line 300, the statement of "R(t_i + \delta t_i) = R(t_i)" does not seems immediate.
- Table 1, the heading for the damage intensity function (the \lambda functions) are hard to read. Some part of the words is also being cut off at the bottom of the fonts.
- Table 1, the legends are being displayed on the next page (line 334) rather than right after the legend headings (line 333).
In conclusion, the paper addresses a grounded issue in identifying the formation of defects int through modelling their corresponding reliability for their intended functions. The method also demonstrates the needs of recognising the multi-criteria of the mechanism of defects formation.
Author Response
Dear Mr. Kongpat Jirapisaisuk, Assistant Editor – MDPI,
Dear Editorial Board of Applied Sciences,
Thank you for the review of our paper entitled “Intensity of the Formation of Defects in Residential Buildings with Regards to Changes in Their Reliability” to be published in the Journal of “Applied Sciences”.
First of all, we appreciate your thoughtful and accurate comments presented by the reviewers. We have carefully considered all remarks and have now completed the revisions incorporating their suggestions in the revised manuscript (copy attached).
We hope that we have taken all your critical and constructive comments into account in the revised paper. We also hope that the current version meets your expectations.
Kind regards,
Jarosław Konior, Marek Sawicki and Mariusz Szóstak
Department of Building Engineering, Faculty of Civil Engineering, Wroclaw University of Science and Technology, 50-370 Wrocław, Poland
Here are answers to reviewer’s comments – one by one:
Detail Comments on the article “Intensity of the Formation of Defects in Residential Buildings with Regards to Changes in Their Reliability” Applied Sciences
General Overview:
The authors performed an exploratory study on the formation of defects by modelling the reliability function of buildings. In particular, the proposing model is tested against a representative group of residential buildings with regards to technical wear. The study used a probabilistic approach, based on the Bayesian conditional variable framework, to model the distribution of damage intensity function. Such a function considered multiple factors such as the building's fulfilment functions and working conditions, as the technical wear of a building is naturally affected by multiple criteria. In addition, a statistical form of the reliability measure is also presented. The study has drawn the conclusion of age not being the main factor that determines the course of the technical wear. Overall, the result of the paper presented a methodology that was able to model the cause-effect relationships of damage to residential buildings.
The problem that this paper address is grounded and the proposed method is logically sound. Moreover, the proposed method addresses the issue with a probabilistic approach in modelling the failure intensity function. The study of the damage intensity function allows one to assess the wear process based on samples obtained from residential buildings.
The following are some comments on the formulation of the proposing method:
Comment 1. In line 237, the reliability function R(t) is said to "assumes values ... where R(0) = 1, and R(\infty) = 0". The statement would be more appropriate to be written as the limit of: lim_{t->\infty} R(t) = 0.
Answer 1. The statement has been supplemented by approach of limit as suggested.
Comment 2. It is stated in line 237 that R(0) = 1. However, the subsequent lines in 246-248 then begins to generalise and define that R(0) is the initial reliability where eq3 stated R(0) = P{Z(\omega\varphi,X,t)}. It is not until line 254 that an additional assumption is stated of saying the event Z(\omega\varphi,X,t) is assumed to be certain and hence R(0) = 1. Perhaps the ordering of formulation should be rearranged.
Answer 2. Yes, this would be more sound and comprehensive. Ordering of formulation has been rearranged.
Comment 3. In line 240 (and 410), the phrase "right-hand time interval" implies there is an intrinsic direction of positive and negative numbers, which is not true—referring the interval as the positive direction would be sufficient.
Answer 3. The interval has been rereferred as the positive direction only without indication of right-hand time.
Comment 4. Eq 2 seems to be incorrect. Based on eq1 and the text descrption, perheps it was supposed to be: F(t) = P{ \tau <= t } = 1 - R(t)?
Answer 4. Eq 2 is correct as F(t)=P{τ>t}=1-R(t)
Comment 5. In some instances ti (without subscript) is used (e.g. line 251) while in some other instance, t_i (as a subscript) is used (e.g. eq5, line265).
Answer 5. This is not ti with subscript as ti_. This is ti less or equal t 0<ti£t (e.g. eq5, line265)
Comment 6. R(t) takes time as the input. But in eq 4, the reliability function R(.) is the first instance of R where it takes the tuple of (\omega\varphi,X,t). Was it meant to be P{Z(\omega\varphi,X,t)}?
Answer 6. Yes, it meant to be P{Z(ωφ,χ,t)} as in eq 3.
Comment 7. Eq8 has an open parenthesis but with no closing parenthesis.
Answer 7. Closing parenthesis inserted.
Comment 8. Given the definition of inequality of t_i in line 251 and in eq6, i.e. 0 < t_i <= t, the statement in line 265 is invalid "in extreme cases for t_i=0 and t_i=\infty". t_i does not takes the value of 0.
Answer 8. This is not stated in the paper as in comment 8. When then in extreme cases for ti=0 and ti=¥, the reliability function takes the values R(0) = 1 and R(∞) = 0, not the other way round.
Comment 9. In line 296, the inverval does not seems correct. Was it meant to be "....in the interval [t_i, t_i + \delta t_i]....."?
Comment 10. In addition to the last comment, does it matter to the analysis if the interval is an open or closed interval (since the two intervals coincide at t_i)?
Answers 9 & 10. This is correct and it does matter to the analysis. When considering any two renovation intervals it is assumed that in the interval [0,ti] no damage was found, and in the interval [ti,ti+Dti] damage could occur, the expression R(ti+Dti)/R(ti) is called the conditional probability of such an event, which assumes that there will be no damage in the interval [ti+Dti] if there is no damage in the interval [0,ti].
Comment 11. Eq12. It is confusing and unclear how the authors derive eq 12 when R is being reformulated under the Bayesian conditional probability. Firstly, with conditional probability, P(A|B) is normally used to denote the probability of A given B. Moreover, the classical Bayes' theorem on conditional probability is in the form of P(A|B) = P(B|A) P(A) / P(B). It is not immediate to the reviewer how eq12 is derived under the Bayes' theorem.
Answer 11. Eq 12 thoroughly correspond to Bayesian formula on conditional probability of A when B is given. R(ti+Dti)/R(ti) = P(A/B) is called the conditional probability of such an event, which assumes that there will be no damage in the interval [ti+Dti] = A if there is no damage in the interval [0,ti] = B.
Comment 12. Line 300, the statement of "R(t_i + \delta t_i) = R(t_i)" does not seems immediate.
Answer 12. The statement is not immediate actually. In very particular, extreme cases, damage may occur at the moment of ti+Dti, and then the probability that is determined by equation (12) will assume the value 1 (because R(ti+Dti)=R(ti)). If the damage occurs at the moment of ti, then P{ti;ti+Dti}=0 (and then R(ti+Dti)=0). Formula (12) indicates the conditional probability of the duration of the correct and fault-free operation of the object.
Comment 13. Table 1, the heading for the damage intensity function (the \lambda functions) are hard to read. Some part of the words is also being cut off at the bottom of the fonts.
Answer 13. Table 1 has been improved. We hope that in the current version is readable.
Comment 14. Table 1, the legends are being displayed on the next page (line 334) rather than right after the legend headings (line 333).
Answer 14. The legend with the heading has been moved right below the table 1 in one line.
General answer to comments 1, 2, 4, 5, 6, 8, 9, 10, 12: Equations citied in comments appear in on-line-writing with descriptive symbols like “omega” instead of original symbol . Therefore, it is difficult to discuss with the reviewer 2 as some meaning may not be comprehensive to the authors. As to avoid any misunderstanding of mathematical transformations we are ready to delete all interim formulas and leave only the final ones (16) and (29) which are enough sufficient basis for the research and conclusions.
In conclusion, the paper addresses a grounded issue in identifying the formation of defects int through modelling their corresponding reliability for their intended functions. The method also demonstrates the needs of recognising the multi-criteria of the mechanism of defects formation.

Round 2
Reviewer 1 Report
The Authors have done the improvement of their work as per the recommendation.
Therefore, this may be considered for the publication now.
